# Repeatability of automated body composition measurement on low dose chest CT in male subjects

Stijn Bunk[1]*, Edwin Bennink[1], Grigory Sidorenkov[2], Hester Gietema[3], Harry Groen[2], Geertruida de Bock[2], Pim A. de Jong[1], Rozemarijn Vliegenthart[2], Firdaus Mohamed Hoesein[1], NELSON-POP consortium[¶]

1 University Medical Center Utrecht, Utrecht University, Utrecht, The Netherlands, 2 University Medical Center Groningen, University of Groningen, Groningen, The Netherlands, 3 Maastricht University Medical Center, Maastricht University, Maastricht, The Netherlands

¶ Membership of the NELSON-POP consortium is provided in the Acknowledgements.
* s.a.o.bunk-2@umcutrecht.nl

## Abstract

### Background

The objective was to determine the most repeatable of three automated body composition methods applied to baseline and short-term follow-up chest CT scans.

### Methods

Areas of skeletal muscle and subcutaneous adipose tissue (SAT) were analyzed in a 1 mm slice close to the aortic arch in a subset of males from the NELSON lung cancer screening trial with baseline and 3–4 month repeat CT scan. We compared three pre-existing machine learning methods we call: *truncated field of view* (FOV), *compensated FOV*, and *extended FOV*, of which the last two can deal with non-overlapping FOV in scans. Repeatability was assessed using Bland-Altman plots and paired T-tests.

### Results

Of 562 males the median (interquartile range) age was 60.8 (56.3–64.8) years. Mean skeletal muscle areas were similar for *truncated (*212 cm²) and *extended FOV (*211 cm²), and slightly lower for *compensated FOV (*208 cm²) ($p < 0.001$). SAT areas were higher with *extended FOV* (156 cm²) compared to *truncated* (132 cm²) and *compensated* FOV (125 cm²) ($p < 0.001$). A small systematic longitudinal difference in skeletal muscle was observed for *extended FOV* (mean±SD $1.7 \pm 17.3$ cm², $p = 0.017$). Limits of agreement for skeletal muscle area were −18.9% to 20.4% for *truncated FOV*, −11.1% to 11.6% for *compensated FOV*, and −16.7% to 18.2% for *extended FOV*.

**Data availability statement:** The Data cannot be shared publicly due to legal restrictions because the data contains potentially identifying patient information. Data are available from the NELSON Data Access Board (contact via https://umcgresearchdatacatalogue.nl/all/resources/NELSON) for researchers who meet the criteria for use of the data. The NELSON Data Access Board will determine whether the intended use of the data matches the original informed consent of the participants of the NELSON trial.

**Funding:** This work was supported by a funding grant from the Dutch Cancer Society (Project Number 9037 to RV), Siemens Healthineers, and the Ministry of Economic Affairs and Climate Policy by means of the Public–Private Partnerships Allowance made available by the Top Sector Life Sciences & Health to stimulate public–private partnerships. This grant also supported the PhD project of SB. The funders had no role in study design, data collection and analysis, decision to publish, or preparation of the manuscript.

**Competing interests:** The authors have read the journal's policy and have the following competing interests: RV declares receiving institutional grants and speaker fees from the following company: Siemens Healthineers. This does not alter our adherence to PLOS ONE policies on sharing data and materials. There are no patents, products in development or marketed products associated with this research to declare.

Corresponding values for SAT area were −37.3% to 38.3%,-30.2% to 29.3%, and −29.1% to 29.9%.

## Conclusion

*Extended FOV* had the second-most repeatable measurements and was unaffected by FOV cutoff. *Compensated FOV* was most repeatable, but underestimated SAT.

---

## Introduction

An estimation of body composition based on chest CT has recently been found to be a predictor for lung cancer mortality [1]. Metrics such as the cross-sectional area of subcutaneous adipose tissue (SAT) and skeletal muscle in the CT cross-sectional images, as well as muscle radiodensity have been shown to be associated with survival in non-small cell lung cancer [2]. Muscle quantity and quality (low radiodensity implying more fat) can predict the progression of lung cancer and overall survival [3,4]. Body mass index (BMI) has also been shown to be linked to overall survival [5,6], but BMI has a weaker predictive value. Furthermore, a reduction over time in skeletal muscle area has been shown to be linked to the disease progression in other cancers such as colorectal cancer [7].

Until recently, obtaining segmentations of areas of thoracic fat and muscle from CT scans was time-consuming manual work with significant inter-reader variability, hampering the use in both research and in the clinic. In recent years, automated methods for determining body composition have been developed, allowing measurements of fat and muscle quantity and quality [8–10]. These automated methods make it feasible to perform body composition evaluations in large cohorts. So far, most studies on body composition have used a single time point body composition evaluation to relate to outcomes. Repeated measurements over time may have additional prognostic value and may be of use in intervention studies but usage is currently limited due to lacking repeatability studies – in other words, the question is whether changes over time are due to real changes or measurement variation. This is of particular importance when evaluating body composition on chest CT scans originally acquired in a lung setting. Specifically, in chest CT part of the body (shoulder region, and in case of obesity, part of circumferential chest wall) can be outside the field-of-view (FOV), leading an underestimation of tissue area. This can worsen the limits of agreement, if there is no compensation for the tissue outside the FOV.

The goal of our research was to determine which method provides the most repeatable measurements of skeletal muscles and SAT in longitudinal datasets of low-dose chest CT. We applied three different automated body composition pipelines to a subset of males from the Dutch and Belgium Lung Cancer Screening Trial (NELSON) with short-term follow-up CT [11] and evaluated the repeatability of the resulting body composition measurements.

## Methods

### Study population

This study is part of a follow-up to NELSON called NELSON-POP, aimed at personalized outcome prediction [12]. The NELSON study was a randomized controlled trial to assess the impact of low-dose chest CT screening in high-risk individuals on lung cancer mortality, when compared to no screening. On November 22, 2000, the Health Council of the Netherlands advised the Minister of Health to give permission to start the trial after a positive test of the 'comprehensibility' of the trial information. On December 23, 2003, the Minister of Health of the Netherlands approved randomisation of persons to the NELSON trial. On March 22, 2006, the trial was retrospectively registered in Het Overzicht van Medisch-wetenschappelijk Onderzoek in Nederland (OMON) (NL-OMON22971). Written informed consent was obtained from all participants. Included participants were current or former smokers with a smoking history of >15 cigarettes/day during >25 years or >10 cigarettes/day during >30 years. People with recent prior cancers were never included in the NELSON trial, and subjects who developed cancer left the trial. These and further in- and exclusion criteria have previously been described [11,13,14]. The screening group underwent low-dose inspiratory chest CT-scans. For the current retrospective study we used a subset of participants, scanned in one of the participating centers (University Medical Center Utrecht), with a short-term follow-up scan for indeterminate baseline nodules (50–500 mm3) (Fig 1). This subset was accessed on 2022-06-10 and could not be used to identify individual participants. This subset contained a group of 562 male subjects with two CT scans taken at an interval of 3–4 months. In the NELSON trial, only 13.9% of the available short-term follow-up CT data from University Medical Center Utrecht was from females, inclusion of which would have introduced unintended variability

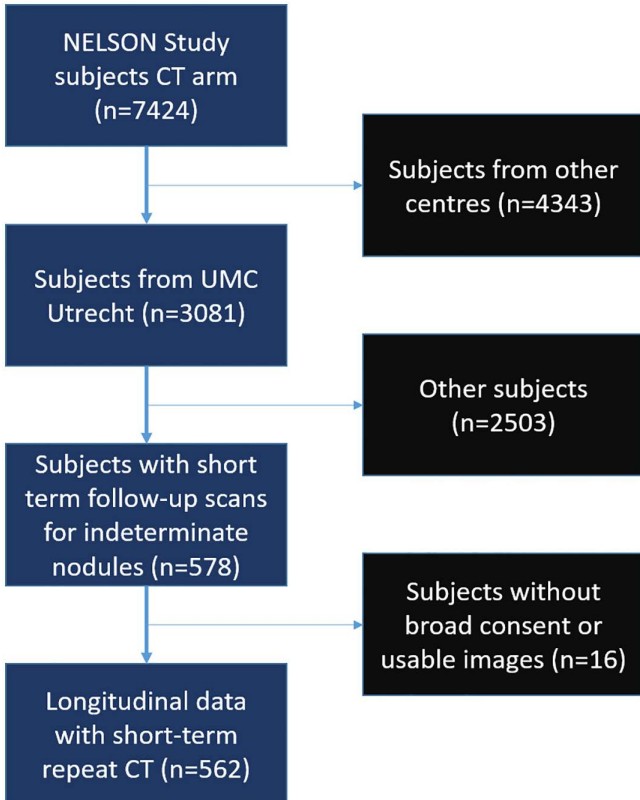

**Fig 1. Flow diagram of data selection.**

in the technical evaluation of the effect of FOV cutoff of tissues on measurement repeatability. Therefore, we only report on male subjects. This subset is appropriate for a repeatability analysis, because in a presumed healthy population body composition is generally expected to be stable over 3–4 months without specific lifestyle programs. Literature on short-term body composition change is extremely limited. One study in 9 persons showed a stable body composition over 3–4 months [15]. Specifically, the subset was used to verify per-subject SAT/skeletal muscle area and density calculation repeatability. Subjects who did not give broad consent for use of data or had no usable images were excluded from the current analysis.

## Image acquisition

Non-contrast enhanced low-dose chest CT scans were obtained with a 16-slice multidetector CT system with collimation of 16 x 0.75 mm (Mx8000 IDT or Brilliance-16P; Philips Healthcare). The following settings were used: fixed 30 mAs and 120 kVp for participants weighing 80 kg or less and 140 kVp for those weighing more than 80 kg. Participants weighing 80 kg or more were scanned with 140 kVp to ensure similar image quality to the participants weighing 80 kg or less, who were scanned with 120 kVp. A standard thorax reconstruction kernel was used to reduce noise (filter B). Data were reconstructed as axial slices of 1 mm at 0.7 mm increment [16,17].

## Body composition methods

Chest CT scans can suffer from FOV cutoff of tissue or FOV changes between scan moments, since the FOV usually does not include the entire body circumference including subcutaneous fat, to optimize image resolution. We are primarily interested in the effect of FOV on repeatability. We will refer to measurement of body composition without compensating for FOV changes as *truncated FOV*.

One option to compensate for measurement of FOV change in longitudinal datasets, is to only include the overlapping areas in the analysis. However, this can cause the area to be analyzed to become substantially smaller. We call this method *compensated FOV*.

Another option is to use a method capable of synthetically extending the FOV, which allows measurement of the full cross-section of the body [18]. We call this method *extended FOV*.

In Fig 2 example images and segmentations are shown for each of the three body composition methods used. In Fig 3 the different processing steps for the data are shown.

All three methods calculate muscle radiodensity by calculating the sum of the HU values of all pixels segmented as muscle divided by the number of those pixels.

## Body composition – Truncated FOV method

The first method is based on previous work by C.P. Bridge concerning an automated segmentation pipeline for thoracic CT scans [8,9,19]. The models for this pipeline are available online (https://github.com/CPBridge/ct_body_composition). This pipeline is based on a two-step approach to segmentation. The first step consists of a convolutional neural network (DenseNet) which selects an appropriate transverse slice from the 3D CT scan, in this case just above the aortic arch. The second step is a convolutional neural network (U-Net) that classifies pixels in the selected slice into specific tissue types, in our case skeletal muscle and SAT. This method was originally provided without weights for the neural networks and had to be retrained for the present study. Details on training the slice selection network and segmentation network can be found in S1 Appendix.

## Body composition – Compensated FOV method

The second method appends a step to compensate for FOV truncation differences between scans to the end of the processing pipeline of the *truncated FOV* method. For each subject the CT image pairs were co-registered. Details on the

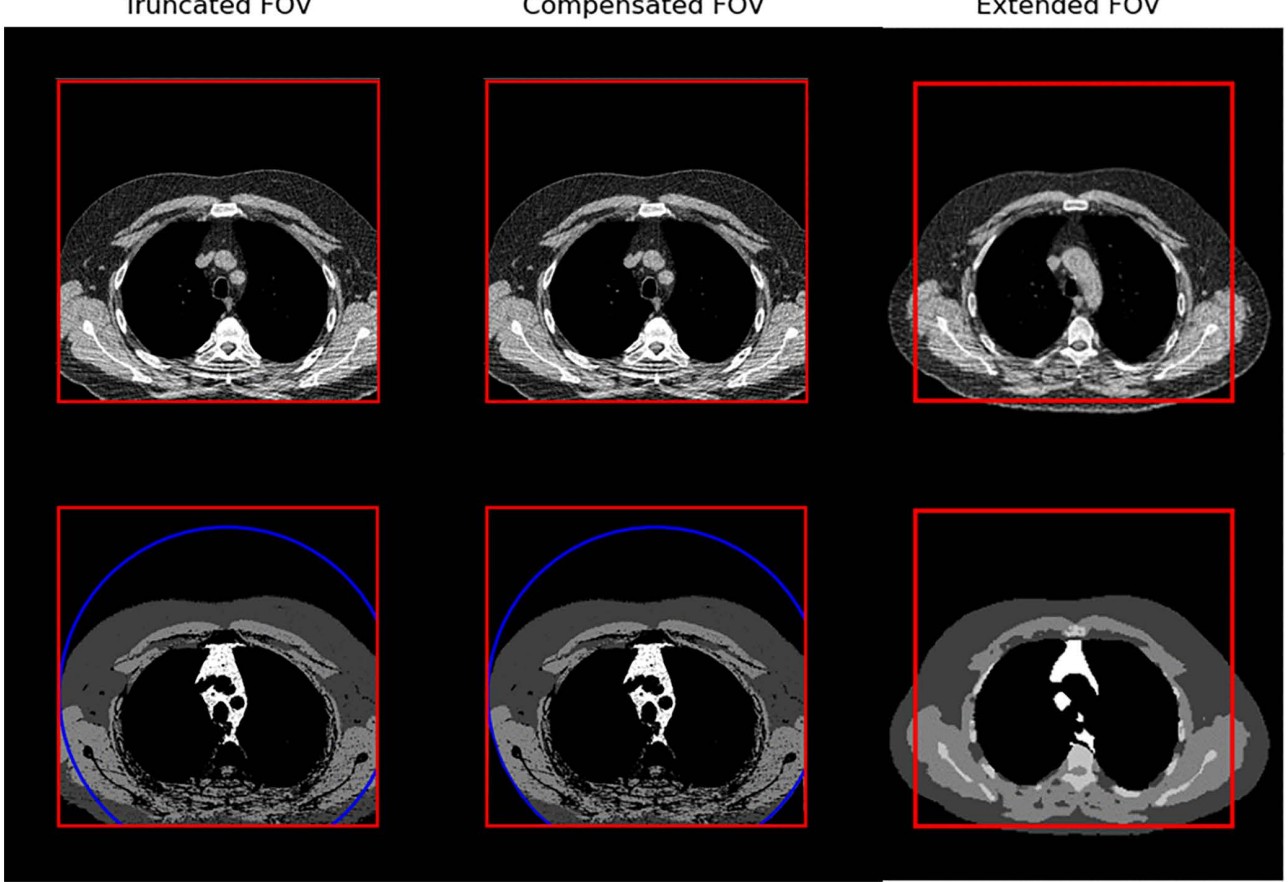

**Fig 2. CT images representing the different methods used.** Red border: edges of originally acquired image. Blue circle: FOV of second image, a follow up CT, acquired for that subject. Dark gray: subcutaneous fat. Medium gray: skeletal muscle.

registration can be found in S3 Appendix. Parts of the image that were not overlapping were excluded in the segmentations. The final area and radiodensity measurements were then calculated based on this cut-off segmentation. The downside of this method is that the amount of available image data is reduced, in particular with multiple repeated CT scans. This effect can be seen schematically in Fig 4.

## Body composition – Extended FOV method

The third method we evaluated was a multi-level FOV extending method called S-EFOV [18,20] (https://github.com/MASILab/S-EFOV). Unlike the *truncated* and *compensated FOV* methods, this *extended FOV* method applies some preprocessing steps. These steps downscale each image slice to 256x256 and clip the HU values to between −150 and +150HU. After these preprocessing steps, the *extended FOV* method first selects three thoracic slices at the T5, T8 and T10 levels, specifically at the transverse processes of each vertebra. We focused on the T5 level, as this most closely matches the level just above the aortic arch measured for the *truncated* and *compensated FOV* methods [21]. Before segmenting the selected slice, the *extended FOV* method uses an image extending model to predict what anatomy would be outside the originally captured FOV and then segments this enlarged FOV. The *extended FOV* method was used without modification. It provides area measurements for skeletal muscle and SAT, as well as metrics for truncation severity

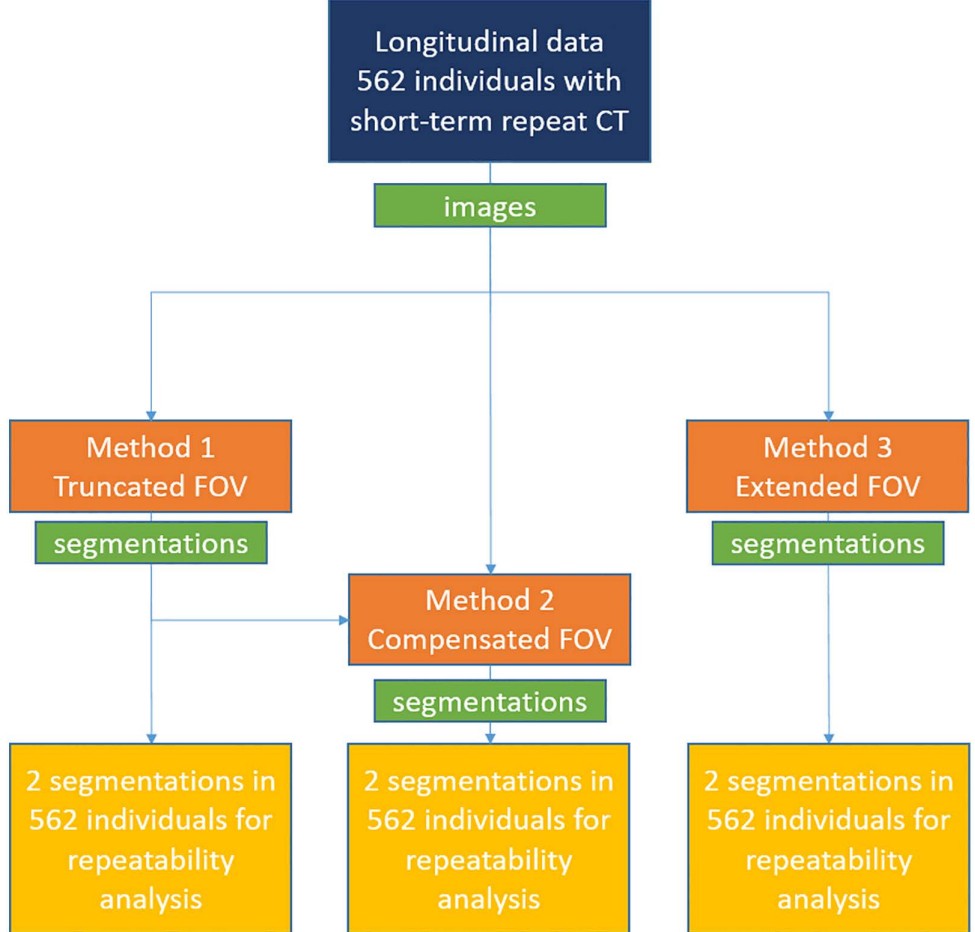

**Fig 3. Flowchart of datasets and methods used.** Method 2 receives both images and segmentations because it appends an image registration step to Method 1. Method 1 and 2 measures just above the aortic arch, Method 3 at thoracic vertebra 5, 8, and 10.

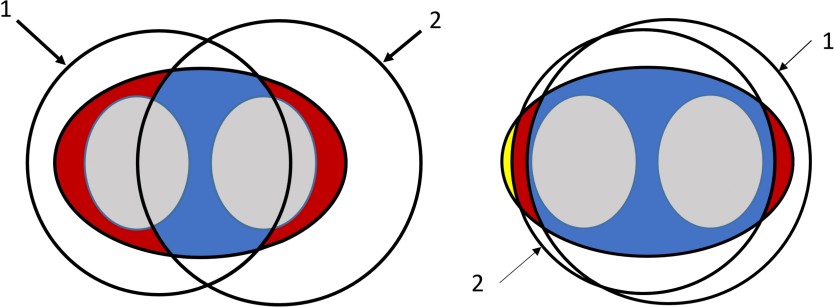

**Fig 4. Schematic example of extreme difference of two scan FOVs and their overlapping area.** The schematic on the left is an extreme example of shifted FOV between scans, the schematic on the right is a more typical example. Gray: lungs. Red: excluded area. Blue: included area. Yellow: area not in either scan. 1: FOV of first scan. 2: FOV of second scan.

and required extension. The truncation severity was defined using the Tissue Cropping Index, which is a measure of the proportion between artificial body boundary caused by FOV truncation and the entire body boundary. A zero value indicates no truncation and higher non-zero values indicate more severe truncation. It is an extension of previous work by the authors of the *extended FOV* method [20]. Unlike the *compensated FOV* method, the *extended FOV* method works on individual CT scans and does not need paired CT scans. The authors report a slice selection Root Square Mean Error of 7.1 (95% CI: [6.0, 8.4]) mm and 7.3 (95% CI: [6.5, 8.2]) mm on the Vanderbilt Lung Screening Program and National Lung Screening Trial test cohort, respectively. Additionally, they report a Dice similarity score of $0.98 \pm 0.02$ for skeletal muscle and $0.99 \pm 0.02$ for SAT at T5 [20].

## Statistical analysis

To test whether the three methods provide repeatable measurements, the data were analyzed qualitatively and quantitatively. The quantitative analysis used Bland-Altman plots to evaluate the repeatability of measurements of the skeletal muscle or SAT tissue areas. $\pm 1.96$SD was used to indicate the limits of agreement. Scans outside of this interval have a 95% chance of being different. Tighter limits of agreement show lower measurement variance and therefore better repeatability. Additionally, a paired samples T-test was used to verify that no significant absolute differences in tissue area or radiodensity existed between each method's two measurements per subject. Paired samples T-tests were also used to determine if differences in measurements existed between methods. We also calculated the Pearson coefficients for the correlations between BMI at baseline and measured SAT area on the first scan for subjects with available BMI data, as an additional verification of measured SAT.

To verify model performance on our data, we performed a qualitative analysis consisting of observing, per tissue type, the 50 subjects with the largest relative disagreement in area between scan moments. For these subjects both scans were visually evaluated side-by-side to determine whether the cause of the disagreement was the result of a method error, and if it was a method error, the size of the error as well. Small errors mean that the segmentations are only off by at most an estimated 10% of total area. Large errors are above this threshold. Using this analysis, the worst-case error rates were determined. The rest of the cohort should have lower error rates.

## Results

### Characteristics of the study population

The longitudinal dataset consisted of 562 subjects with a median age of 60.8 (25%−75%, 56.3–64.8) years. All participants were male. For 247 subjects the BMI was available (mean±SD, $25.7 \pm 3.4$).

### Error rates in segmentations

The severity distribution of percentages of segmentation errors for the 50 scans with the largest area disagreement between scans is lowest for the extended FOV (Table 1). Across all the longitudinal scans, the Tissue Cropping Index at T5 was $0.26 \pm 0.12$ (mean±SD).

### Repeatability of measurements

The measured areas of skeletal muscle and SAT were normally distributed for all three methods. There were clear differences in the repeatability characteristics between the *truncated*, c*ompensated*, and e*xtended FOV* methods (Table 2). *Compensated FOV* had the tightest limits of agreement, followed by *extended FOV*, with *truncated FOV* having the widest limits. In Fig 5, the Bland-Altman plots of the relative difference in area between the first and second scan for each of the methods and tissue types are shown. Based on the paired samples T-test, only the *extended FOV* method had a significant difference ($p = 0.017$) in measured area for skeletal muscle between the first and second scan, but this difference

**Table 1. Percentages of scans with a segmentation error in the 50 subjects with the largest errors.**

|  | Truncated FOV | | Compensated FOV | | Extended FOV | |
|---|---|---|---|---|---|---|
| Error Severity | Skeletal Muscle | SAT | Skeletal Muscle | SAT | Skeletal Muscle | SAT |
| None (%) | 38 | 56 | 30 | 32 | 58 | 72 |
| Small (%) | 44 | 30 | 50 | 38 | 32 | 24 |
| Large (%) | 18 | 14 | 20 | 30 | 8 | 4 |

None = no error. Small = less than 10% of area error. Large = more than 10% of area error.

**Table 2. Measurement data of the 562 subjects for the *Truncated*, *Compensated*, and *Extended FOV* methods.**

|  |  | Mean Area (cm²) | Mean Difference (%) | Lower limit (%) | Upper Limit (%) | Agreement *p*-value | ICC |
|---|---|---|---|---|---|---|---|
| Truncated FOV | Skeletal Muscle | 212 | 0.7 | −18.9 | 20.4 | 0.125 | 0.914 [0.898, 0.927] |
|  | SAT | 132 | 0.5 | −37.3 | 38.3 | 0.806 | 0.967 [0.962, 0.972] |
| Compensated FOV | Skeletal Muscle | 208 | 0.3 | −11.1 | 11.6 | 0.421 | 0.973 [0.968, 0.977] |
|  | SAT | 125 | −0.4 | −30.2 | 29.3 | 0.126 | 0.987 [0.984, 0.989] |
| Extended FOV | Skeletal Muscle T5 | 211 | 0.7 | −16.7 | 18.2 | 0.017 | 0.942 [0.932, 0.951] |
|  | SAT T5 | 156 | 0.4 | −29.1 | 29.9 | 0.995 | 0.984 [0.981, 0.986] |

SAT = Subcutaneous Adipose Tissue. Upper and lower limit indicate the 95% agreement boundaries, with the value being the percentage difference between a subject's two scans relative to the mean. Agreement *p*-value = Paired samples T-test *p*-value of area measured by method in first scan compared to second scan. ICC = intraclass correlation coefficient, values in square brackets are the 95% confidence interval.

was small (mean±SD 1.7 ± 17.3 cm²). For SAT none of the models had a significant difference between the first and second scan. Mean skeletal muscle areas were similar *for truncated FOV (*212 cm²), and *extended FOV (*211 cm²), but was slightly lower for *compensated FOV (*208 cm²) (p < 0.001). Additionally, SAT areas were higher with *extended FOV* (156 cm²) compared to *truncated FOV* (132 cm²) and *compensated FOV* (125 cm²) (*p* < 0.001). Finally, subjects with the same scan kVp for both scans had slightly more repeatable measurements versus those subjects with different scan kVp values. For details, see Table A in S4 Appendix.

### Correlation of fat area with BMI

For each of the three methods, the correlation of SAT with BMI was evaluated. The Pearson correlations of subcutaneous fat with baseline BMI using the *truncated FOV* method were 0.802 (p < 0.001) for the first scan moment and 0.822 (*p* < 0.001) for the second scan moment. For the *compensated FOV* method they were 0.801 (*p* < 0.001) and 0.815 (*p* < 0.001), respectively. For the *extended FOV* method, they were 0.829 (P < 0.001) and 0.823 (*p* < 0.001), respectively. This suggests that the extended FOV method did not introduce erroneous tissue, but actually slightly improved the association with BMI. Scatterplots of these correlations are shown in S2 Fig.

### Radiodensity measurements

The measured radiodensities of the tissues showed a discrepancy between the *extended FOV* method and *truncated/ compensated FOV* methods. *Extended FOV* measured a mean muscle radiodensity across all scans of 25.2HU, versus 37.8HU for *truncated FOV* and 37.7HU for *compensated FOV*. More details are shown in Table 3. As expected, we found that scans collected at 120 kVp had a slightly higher measured radiodensity compared to those collected at 140 kVp. For details, see Table B in S4 Appendix.

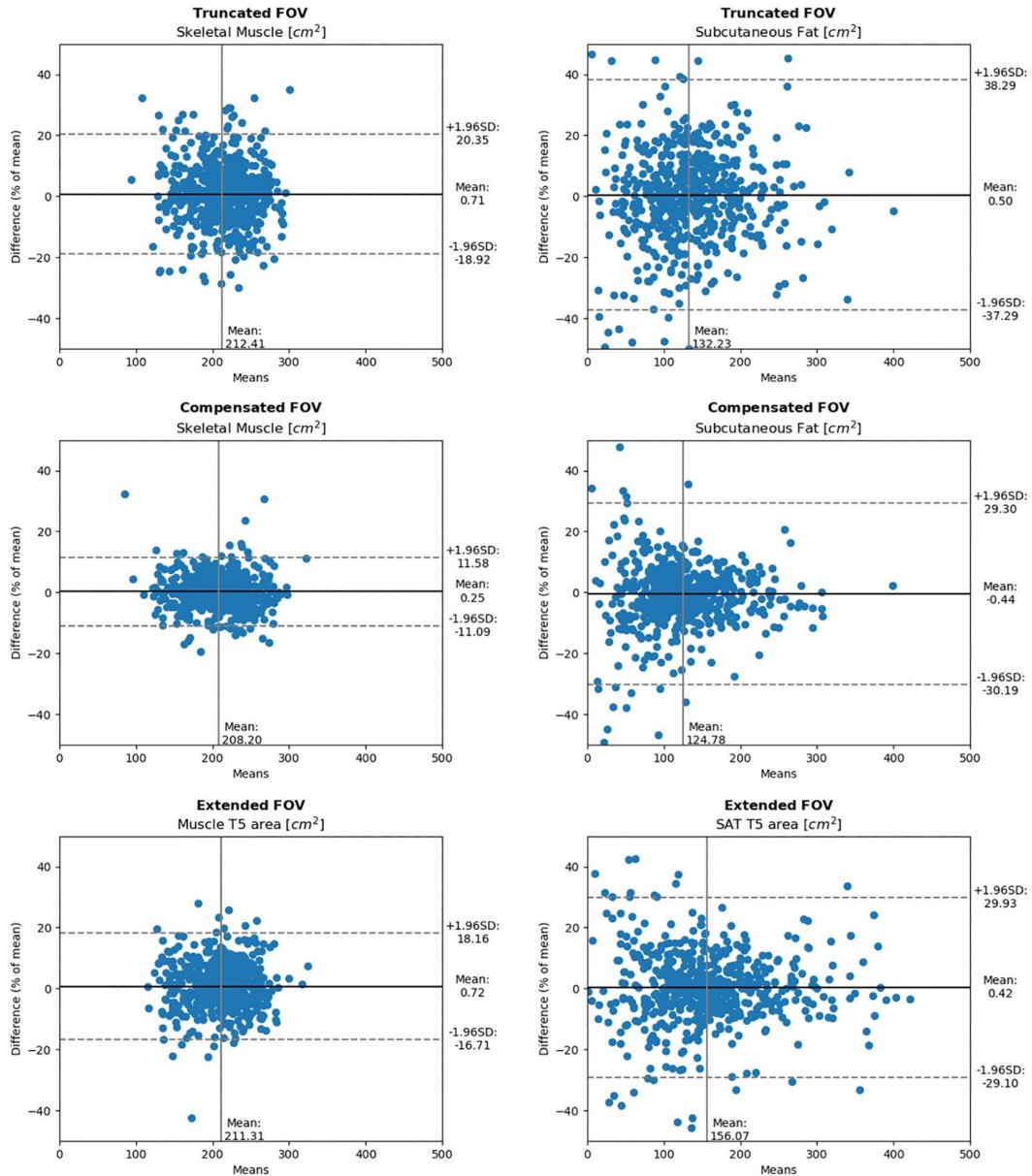

**Fig 5. Bland-Altman plots of difference in area between two scans.** The difference is displayed as a percentage of the mean area of those two scans. The vertical mean line is mean area in cm² across all scans. The horizontal mean line is the mean difference between two scans.

## Discussion

We investigated the longitudinal repeatability of three different body composition calculation methods, *truncated, compensated* and *extended FOV*. First, we observed that the *truncated* and *compensated FOV* methods underestimated the actual area of SAT, due to the scan FOV excluding some tissue. There were very minimal differences in skeletal muscle areas. Secondly, we observed that the *compensated* FOV method had the highest interscan agreement, followed by *extended FOV* and then *truncated FOV.* Thirdly, We also observed a systematic difference in measured tissue

**Table 3. Mean muscle radiodensities for the 562 subjects as measured by the *truncated, compensated* and *extended FOV* methods.**

| | | First Scan Moment | p-value relative to | | | Second Scan Moment | p-value relative to | | | p-value |
|---|---|---|---|---|---|---|---|---|---|---|
| | | | Truncated FOV | Compensated FOV | Extended FOV | | Truncated FOV | Compensated FOV | Extended FOV | |
| Truncated FOV | Muscle Mean (HU) | 38.0±4.5 | N/A | <0.001 | <0.001 | 37.6±4.4 | N/A | <0.001 | <0.001 | 0.027 |
| | Muscle SD Mean (HU) | 33 | N/A | <0.001 | <0.001 | 33.0 | N/A | <0.001 | <0.001 | 0.115 |
| Compensated FOV | Muscle Mean (HU) | 37.9±4.4 | <0.001 | N/A | <0.001 | 37.5±4.4 | <0.001 | N/A | <0.001 | 0.025 |
| | Muscle SD Mean (HU) | 34 | <0.001 | N/A | <0.001 | 34.0 | <0.001 | N/A | <0.001 | 0.642 |
| Extended FOV | Muscle Mean (HU) | 25.4±5.2 | <0.001 | <0.001 | N/A | 25.0±5.3 | <0.001 | <0.001 | N/A | 0.019 |
| | Muscle SD Mean (HU) | 39 | <0.001 | <0.001 | N/A | 39.5 | <0.001 | <0.001 | N/A | 0.487 |

HU = Hounsfield Units; SD = Standard Deviation. Muscle Mean reported ± SD. Muscle SD Mean refers to the mean of all scans' standard deviation. The 'p-value relative to' columns compare the HU values to those of the other methods using a paired-samples T-test. N/A indicates that HU values of methods' were not compared to themselves. The 'p-value' column compares the HU measured during the first scan moment to the HU measured during the second scan moment using a paired samples T-test.

radiodensity between the *truncated* and *compensated FOV* methods and the *extended FOV* method. Finally, we observed that the *extended FOV* method had the lowest rate of segmentation errors.

## Evaluation of repeatability metrics of fat and muscle area

The higher upper and lower limits of agreement of the *truncated FOV* method as compared to the *compensated* and *extended FOV* methods indicate that the *truncated FOV* method's measurements are the least repeatable. The *compensated FOV* method had the most repeatable measurements, as indicated by its limits of agreement being the lowest. However, this method has downsides. Specifically, the *compensated FOV* method requires multiple scans of the same subject to compensate for FOV cutoff due to the image registration requirement, and for each additional scan measured the lost area due to non-overlapping FOV exclusion will increase. This reduction in area is indicated by the *compensated FOV* method having the lowest mean area of skeletal muscle and SAT. The *extended FOV* method lacks these downsides. It measures closest to the actual subject's area, which is indicated by it having the highest mean area of skeletal muscle and SAT. *Extended FOV* also has similar limits of agreement for SAT area as compared to *compensated FOV*, and for skeletal muscle has limits of agreement in between *compensated* and *truncated FOV*. *Extended FOV* also has the lowest rate of segmentation errors, increasing confidence in its measurements. Depending on how these factors are prioritized, for a longitudinal analysis of fat and muscle areas on chest CT scans, especially when the absolute value of the measurement is important, the *extended FOV* method is likely the most suited.

To the best of our knowledge, this is the first study on the repeatability of methods performing automated body composition measurements on chest CT. Therefore, we are not able to make a comparison to existing literature.

## Radiodensity discrepancies between the body composition methods

There was a discrepancy between the tissue radiodensity as measured on the area segmented by the *truncated/compensated FOV* methods and the area segmented by the *extended FOV* method. We found that this discrepancy was not caused by preprocessing steps such as a) the *extended FOV* method clipping radiodensity from −150 HU to +150 HU, b) the *extended FOV* method using 256x256 voxel images versus truncated/compensated using 512x512 voxels, or c) differences in the method by which radiodensity is measured inside a tissue's segmented voxels. When the preprocessing steps of the *extended FOV* method named in a) and b) were applied as post-processing steps to the CT scans and segmentations from the *truncated/compensated FOV* method, the radiodensity measurements provided by the *truncated/*

compensated FOV method did not change. The difference in mean measured density in HU seems to stem from differences in their segmentations. The *truncated/compensated FOV* method much more aggressively excludes pixels which were affected by things like beam hardening. An example of this can be seen in Fig 6, showing the same scan as processed by both methods. The *truncated FOV* segmentation contains many more holes than the *extended FOV* segmentation. As each method is internally consistent, we do not expect the differences in radiodensity measurements between methods to have any clinical implications, provided any clinical decision thresholds created are calibrated to each method individually.

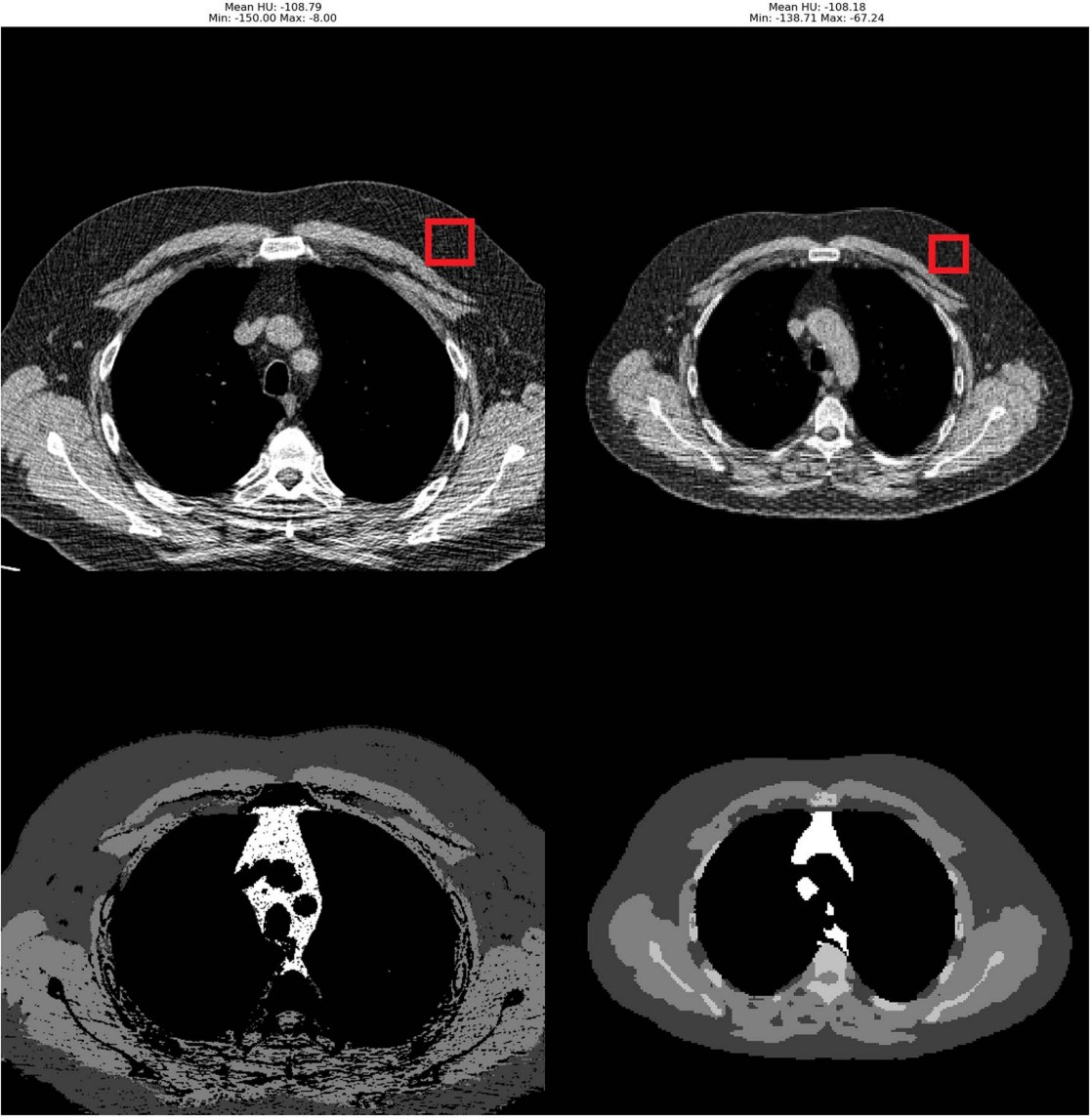

**Fig 6. Radiodensity evaluation between truncated FOV on the left and extended FOV on the right.** The radiodensity is evaluated in the areas marked by the red box in the top right of each method. The truncated FOV method had a mean density in the selected area of −108.79HU and the extended FOV method had a mean density of −108.18HU.

### Relevance for chest disease

Having access to an automated method for determining longitudinal changes in fat and muscle volume can allow for longitudinal analyses in large cohorts. For example, the change in fat and muscle volume over time could be analyzed in relation to overall survival, lung cancer incidence, and lung cancer survival. Additionally, it may allow for the detection of muscle loss during treatment of (lung) cancer patients. Patients with rapid muscle loss could benefit from modifications to their treatment plans to reduce this loss. Furthermore, the measurements could be used to assess the effect of pre-habilitation for major interventions or speed of recovery between treatment cycles. There could also be applications in the treatment and management of diseases like emphysema or chronic bronchitis, which are related to cachexia and fat accumulation.

For implementation into clinical workflows the time required by each method to perform its measurements is also of interest. Sometimes there is a trade-off between computational time and accuracy, where high performing methods require longer to perform the analysis. However, none of the three methods we evaluated differed significantly in the time required to perform the measurements. The *compensated FOV* method required more computation than the *truncated FOV* method due to the need to also register the images, but this additional time cost was negligible. The *truncated FOV* method and *extended FOV* method required similar amounts of time to process data, despite the *extended FOV* method performing the additional step of FOV extension. For all methods, averaged across all scans, processing a single scan took less than a minute.

### Strengths and limitations

Our study had a number of important strengths. Firstly, we had a large subject sample, enhancing the reliability of the results. Secondly, we evaluated a variety of different approaches to this problem, enhancing the understanding of the current capabilities of body composition segmentation models. Lastly, all the methods evaluated are freely available and can thus be used by further studies.

Our study also has some limitations. Firstly, our study group comprises only men, which means the results may not be applicable to female subjects, where anatomical differences in the chest can distort segmentations. Secondly, we assume no systematic changes in body composition over a 3–4 month interval, as lifestyle programs were no part of the screening trial (outside of advice to stop smoking). However, this assumption may not universally hold in other populations, especially those in which interventions are undertaken. Nevertheless, our results may underestimate repeatability, which could be slightly better with shorter scan interval. Thirdly, though care was taken to compare similar levels in the chest, the two methods did not compare the exact same slice, possibly reducing the strength of the comparison between methods. However, it is unlikely this affected the comparisons within methods. Fourthly, our participants had indeterminate nodules. However, although we did observe a mean change (bias) in in tissue area between the two scan moments for *extended FOV*, this difference was very small, making it unlikely this affected the results. Lastly, we only looked at the correlation between SAT and BMI in a subset of subjects. We don't expect that this majorly affected the results.

### Conclusion

Automated body composition methods with and without FOV correction, showed similar values for skeletal muscle areas, but impacted subcutaneous adipose tissue areas. *Compensated FOV*, when evaluated purely on the wideness of its limits of agreement, has the best repeatability. However, it suffers from reduced measurement areas because it excludes non-overlapping areas of longitudinal scans. This reduction in area will only increase in severity if more than two longitudinal scans are evaluated. *Extended FOV* had the second-most repeatable measurements, and by reconstructing areas excluded on the original image it provides measurements closest to the actual subject area, had their chest been fully imaged.

## Supporting information

**S1 Appendix. Training of Truncated FOV method's models.**
(DOCX)

**S2 Fig. Correlation of BMI with SAT.** Correlation plots between BMI and SAT for the truncated, compensated, and extended FOV methods.
(TIF)

**S3 Appendix. Details on the registration process for *compensated FOV*.**
(DOCX)

**S4 Appendix. Differences in measurements based on scan kVp.**
(DOCX)

## Acknowledgments

All non-author members of the NELSON-POP consortium:

From University Medical Center Rotterdam, Erasmus University Rotterdam, Rotterdam, The Netherlands: Joachim G. Aerts, Robin Cornelissen, Ralph Stadhouders, Jeroen G.J. van Rooij, Lianne Trap.

From University Hospital Leuven, Catholic University of Leuven, Leuven, Belgium: Kristiaan Nackaerts, Walter de Wever.

From Radboud University Medical Center, Nijmegen, The Netherlands: Mathias Prokop, Cornelia Schaefer-Prokop, Colin Jacobs, Noa Antonissen.

From University Medical Center Groningen, University of Groningen, Groningen, The Netherlands: Marjolein A. Heuvelmans, Danrong Zhong, Nils van der Velden, Thijs Bruins-Slot.

From University Medical Center Utrecht, Utrecht University, Utrecht, The Netherlands: George S. Downward, Roel C.H. Vermeulen.

## Author contributions

**Conceptualization:** Stijn Bunk, Edwin Bennink, Pim A de Jong, Rozemarijn Vliegenthart, Firdaus Mohamed Hoesein.

**Data curation:** Edwin Bennink.

**Formal analysis:** Stijn Bunk.

**Funding acquisition:** Rozemarijn Vliegenthart.

**Methodology:** Stijn Bunk, Edwin Bennink, Pim A de Jong, Rozemarijn Vliegenthart, Firdaus Mohamed Hoesein.

**Resources:** Edwin Bennink.

**Supervision:** Edwin Bennink, Pim A de Jong, Rozemarijn Vliegenthart, Firdaus Mohamed Hoesein.

**Validation:** Grigory Sidorenkov.

**Writing – original draft:** Stijn Bunk.

**Writing – review & editing:** Edwin Bennink, Grigory Sidorenkov, Hester Gietema, Harry Groen, Geertruida de Bock, Pim A de Jong, Rozemarijn Vliegenthart, Firdaus Mohamed Hoesein.

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
