## [Decision Letter · Decision Letter 0]

21 Jan 2026

Dear Dr. Bunk,

Thank you for submitting your manuscript to PLOS ONE. After careful consideration, we feel that it has merit but does not fully meet PLOS ONE’s publication criteria as it currently stands. A few critical comments from the reviewers need to be addressed and some details need to be added. Therefore, we invite you to submit a revised version of the manuscript that addresses the points raised during the review process.

We look forward to receiving your revised manuscript.

Kind regards,

Zhentian Wang, Ph.D.

Academic Editor

PLOS One

**Journal Requirements:**

1. When submitting your revision, we need you to address these additional requirements. Please ensure that your manuscript meets PLOS ONE's style requirements, including those for file naming. The PLOS ONE style templates can be found at https://journals.plos.org/plosone/s/file?id=wjVg/PLOSOne_formatting_sample_main_body.pdf and https://journals.plos.org/plosone/s/file?id=ba62/PLOSOne_formatting_sample_title_authors_affiliations.pdf 2. We note that the grant information you provided in the ‘Funding Information’ and ‘Financial Disclosure’ sections do not match.  When you resubmit, please ensure that you provide the correct grant numbers for the awards you received for your study in the ‘Funding Information’ section. 3. Thank you for stating in your Funding Statement: This work is supported by funding from the Dutch Cancer Society, Siemens Healthineers, and by the Ministry of Economic Affairs and Climate Policy by means of the Public–Private Partnerships Allowance made available by the Top Sector Life Sciences & Health to stimulate public–private partnerships. Funding information for this article has been deposited with the Crossref Funder RegistryRV recieved the grant under project number 9037 from the Dutch Cancer Society.https://www.kwf.nl/onderzoek/onderzoeksdatabase/longkankerscreening-op-maat-met-multidata-voorspelmodellenThe funders had no role in study design, data collection and analysis, decision to publish, or preparation of the manuscript.  Please provide an amended statement that declares *all* the funding or sources of support (whether external or internal to your organization) received during this study, as detailed online in our guide for authors at http://journals.plos.org/plosone/s/submit-now.  Please also include the statement “There was no additional external funding received for this study.” in your updated Funding Statement. Please include your amended Funding Statement within your cover letter. We will change the online submission form on your behalf. 4. Thank you for stating the following in the Competing Interests section: I have read the journal's policy and the authors of this manuscript have the following competing interests:RV declares a relationship with the following company: Siemens HealthineersThe other authors of this manuscript declare no relationships with any companies, whose products or services may be related to the subject matter of the article.  We note that one or more of the authors are employed by a commercial company: Siemens Healthineers.  a. Please provide an amended Funding Statement declaring this commercial affiliation, as well as a statement regarding the Role of Funders in your study. If the funding organization did not play a role in the study design, data collection and analysis, decision to publish, or preparation of the manuscript and only provided financial support in the form of authors' salaries and/or research materials, please review your statements relating to the author contributions, and ensure you have specifically and accurately indicated the role(s) that these authors had in your study. You can update author roles in the Author Contributions section of the online submission form. Please also include the following statement within your amended Funding Statement. “The funder provided support in the form of salaries for authors, but did not have any additional role in the study design, data collection and analysis, decision to publish, or preparation of the manuscript. The specific roles of these authors are articulated in the ‘author contributions’ section.”If your commercial affiliation did play a role in your study, please state and explain this role within your updated Funding Statement.  b. Please also provide an updated Competing Interests Statement declaring this commercial affiliation along with any other relevant declarations relating to employment, consultancy, patents, products in development, or marketed products, etc.   Within your Competing Interests Statement, please confirm that this commercial affiliation does not alter your adherence to all PLOS ONE policies on sharing data and materials by including the following statement: "This does not alter our adherence to  PLOS ONE policies on sharing data and materials.” (as detailed online in our guide for authors http://journals.plos.org/plosone/s/competing-interests) . If this adherence statement is not accurate and  there are restrictions on sharing of data and/or materials, please state these. Please note that we cannot proceed with consideration of your article until this information has been declared. Please include both an updated Funding Statement and Competing Interests Statement in your cover letter. We will change the online submission form on your behalf. 5. We note that you have indicated that there are restrictions to data sharing for this study. For studies involving human research participant data or other sensitive data, we encourage authors to share de-identified or anonymized data. However, when data cannot be publicly shared for ethical reasons, we allow authors to make their data sets available upon request. For information on unacceptable data access restrictions, please see http://journals.plos.org/plosone/s/data-availability#loc-unacceptable-data-access-restrictions.  Before we proceed with your manuscript, please address the following prompts: a) If there are ethical or legal restrictions on sharing a de-identified data set, please explain them in detail (e.g., data contain potentially identifying or sensitive patient information, data are owned by a third-party organization, etc.) and who has imposed them (e.g., a Research Ethics Committee or Institutional Review Board, etc.). Please also provide contact information for a data access committee, ethics committee, or other institutional body to which data requests may be sent. b) If there are no restrictions, please upload the minimal anonymized data set necessary to replicate your study findings to a stable, public repository and provide us with the relevant URLs, DOIs, or accession numbers. Please see http://www.bmj.com/content/340/bmj.c181.long for guidelines on how to de-identify and prepare clinical data for publication. For a list of recommended repositories, please see https://journals.plos.org/plosone/s/recommended-repositories. You also have the option of uploading the data as Supporting Information files, but we would recommend depositing data directly to a data repository if possible. Please update your Data Availability statement in the submission form accordingly. 6. One of the noted authors is a group or consortium “NELSON-POP study group”. In addition to naming the author group, please list the individual authors and affiliations within this group in the acknowledgments section of your manuscript. Please also indicate clearly a lead author for this group along with a contact email address. 7. Please amend either the abstract on the online submission form (via Edit Submission) or the abstract in the manuscript so that they are identical. 8. Please upload a new copy of Figures 5, 6, S2, S4 and S5, as the detail is not clear. Please follow the link for more information:  https://journals.plos.org/plosone/s/figures 9. Please include captions for your Supporting Information files at the end of your manuscript, and update any in-text citations to match accordingly. Please see our Supporting Information guidelines for more information: http://journals.plos.org/plosone/s/supporting-information. 10. If the reviewer comments include a recommendation to cite specific previously published works, please review and evaluate these publications to determine whether they are relevant and should be cited. There is no requirement to cite these works unless the editor has indicated otherwise. 

**Additional Editor Comments:**

This study evaluates and compares the repeatability of three automated methods for measuring body composition—specifically skeletal muscle area and subcutaneous adipose tissue area—from low-dose chest CT scans in a longitudinal setting. The manuscript is concise, well-organized, and clearly presented. Although it does not introduce significant technical novelty, it represents a valid investigation that offers meaningful insights for the medical field, supported by a large dataset. That said, I have several technical questions that require clarification:

First, I have concerns regarding the radiodensity discrepancies observed among the three methods. The extended FOV method underwent notably different preprocessing compared to the other two methods. It remains unclear how the authors could confidently claim—and rule out—that these discrepancies were not attributable to preprocessing. In particular, the fact that discrepancies persisted when the extended FOV method was applied to datasets from the other two methods suggests to me that its processing pipeline may introduce systematic errors. As the authors noted, downsampling could indeed lead to segmentation issues.

Second, regarding the study population, all CT data are from male participants. The rationale for including only males—or whether only male data were available—is not explained. Further details on the population selection criteria are needed.

Third, some findings appear obvious and contribute little novelty. For example, in line 222, the statement “We also found that scans collected at 120 kVp … compared to those collected at 140 kVp” reflects a well-understood principle that higher kVp reduces attenuation and thus lowers radiodensity measurements. This finding seems unnecessary and not directly relevant to the core aims of the study.

Finally, certain statements are overly vague. For instance, line 95 states: “Participants with higher weight were scanned with a higher kVp to ensure similar image quality as the participants with lower weight.” However, it is unclear how many participants were affected and what specific kVp values were used in these cases.

Minor formality issues:

Line 207, cm2 is not current format of the unit.

Footnote of Table 2. SAT=Subcutaneous Fat. Please stay consistent, SAT = subcutaneous adipose tissue.

Reviewers' comments:

**Comments to the Author**

1. Is the manuscript technically sound, and do the data support the conclusions?

Reviewer #1: Yes

2. Has the statistical analysis been performed appropriately and rigorously?

Reviewer #1: Yes

3. Have the authors made all data underlying the findings in their manuscript fully available?

Reviewer #1: No

4. Is the manuscript presented in an intelligible fashion and written in standard English?

Reviewer #1: Yes

**Reviewer #1:** The manuscript addresses the repeatability of automated body composition measurements on longitudinal chest CT scans. Specifically, it evaluates three approaches for compensating field-of-view (FOV) changes between scans: truncated FOV, compensated FOV, and extended FOV. Using a substantial cohort of 562 male participants from the NELSON lung cancer screening trial with baseline and short-term follow-up scans, the authors compare these methods for skeletal muscle and subcutaneous adipose tissue (SAT) area measurements. Repeatability was assessed using Bland–Altman plots, paired t-tests, and intraclass correlation coefficients (ICC). The manuscript addresses the repeatability of automated body composition measurements on longitudinal chest CT scans. Specifically, it evaluates three approaches for compensating field-of-view (FOV) changes between scans: truncated FOV, compensated FOV, and extended FOV. Using a substantial cohort of 562 male participants from the NELSON lung cancer screening trial with baseline and short-term follow-up scans, the authors compare these methods for skeletal muscle and subcutaneous adipose tissue (SAT) area measurements. Repeatability was assessed using Bland–Altman plots, paired t-tests, and intraclass correlation coefficients (ICC). The manuscript addresses the repeatability of automated body composition measurements on longitudinal chest CT scans. Specifically, it evaluates three approaches for compensating field-of-view (FOV) changes between scans: truncated FOV, compensated FOV, and extended FOV. Using a substantial cohort of 562 male participants from the NELSON lung cancer screening trial with baseline and short-term follow-up scans, the authors compare these methods for skeletal muscle and subcutaneous adipose tissue (SAT) area measurements. Repeatability was assessed using Bland–Altman plots, paired t-tests, and intraclass correlation coefficients (ICC). The manuscript addresses the repeatability of automated body composition measurements on longitudinal chest CT scans. Specifically, it evaluates three approaches for compensating field-of-view (FOV) changes between scans: truncated FOV, compensated FOV, and extended FOV. Using a substantial cohort of 562 male participants from the NELSON lung cancer screening trial with baseline and short-term follow-up scans, the authors compare these methods for skeletal muscle and subcutaneous adipose tissue (SAT) area measurements. Repeatability was assessed using Bland–Altman plots, paired t-tests, and intraclass correlation coefficients (ICC).

Key findings indicate that the compensated FOV method achieved the highest repeatability but underestimated SAT due to exclusion of non-overlapping regions. Extended FOV produced measurements closest to actual anatomy and exhibited the lowest segmentation error rates, though it introduced a small systematic difference in skeletal muscle area between scans. The authors conclude that extended FOV may be preferable for longitudinal analyses where absolute measurement values are critical.

Concerns:

1. Population Limitation: The study includes only male participants, limiting generalizability. Anatomical differences in women may affect segmentation accuracy. Have the authors considered validating their approach on publicly available datasets that include female subjects (e.g., https://atm22.grand-challenge.org/dataset/)?

2. Assumption of Stability: The assumption of stable body composition over 3–4 months may not universally hold, despite limited supporting evidence cited.

3. Slice Selection Variability: Comparisons between methods are weakened by non-identical slice selection across approaches.

4. Radiodensity Discrepancy: The observed differences in radiodensity measurements between extended FOV and other methods are noted but not fully explained, leaving uncertainty regarding clinical interpretation.

5. Computational Efficiency: The manuscript would benefit from a discussion on computational efficiency and integration into clinical workflows. Specifically, what is the trade-off between computational time and accuracy across the best- and worst-performing methods?

.

Reviewer #1: No

---

## [Author Response · Author response to Decision Letter 1]

2 Feb 2026

Response to Reviewers

We would like to thank the Editor and Reviewer for their careful examination of our manuscript and their valuable feedback. We note the editor and reviewer have several concerns. The following are comment-by-comment replies and excerpts of the changes made. The changes in the excerpts are highlighted in red. We hope the changes to our manuscript satisfy the Editor’s and Reviewer’s concerns.

Editor’s comments:

Author’s response: We have studied the style templates and file naming requirements and updated the manuscript style, references, and file names to match.

Author’s response: Our apologies for incorrectly reporting the grant number. The correct grant number is 9037. This grant number has also been provided in the updated Funding Statement. It may also be important to note that the Dutch Cancer Society is the English name for KWF Kankerbestrijding.

This work is supported by funding from the Dutch Cancer Society, Siemens Healthineers, and by the Ministry of Economic Affairs and Climate Policy by means of the Public–Private Partnerships Allowance made available by the Top Sector Life Sciences & Health to stimulate public–private partnerships. Funding information for this article has been deposited with the Crossref Funder Registry

RV recieved the grant under project number 9037 from the Dutch Cancer Society.

https://www.kwf.nl/onderzoek/onderzoeksdatabase/longkankerscreening-op-maat-met-multidata-voorspelmodellen

Author’s response: Our funding statement declares all funding and sources of support. We have updated our funding statement to indicate the first author’s PhD project was supported by the Dutch Cancer Society grant. We have updated our Funding Statement to include “There was no additional external funding received for this study.”.

This work is supported by funding from the Dutch Cancer Society, Siemens Healthineers, and by the Ministry of Economic Affairs and Climate Policy by means of the Public–Private Partnerships Allowance made available by the Top Sector Life Sciences & Health to stimulate public–private partnerships. Funding information for this article has been deposited with the Crossref Funder Registry.

RV received the indicated grant under project number 9037 from the Dutch Cancer Society. https://www.kwf.nl/onderzoek/onderzoeksdatabase/longkankerscreening-op-maat-met-multidata-voorspelmodellen. The PhD project of the first author was supported by this grant.

There was no additional external funding received for this study.

I have read the journal's policy and the authors of this manuscript have the following competing interests:

RV declares a relationship with the following company: Siemens Healthineers

The other authors of this manuscript declare no relationships with any companies, whose products or services may be related to the subject matter of the article.

We note that one or more of the authors are employed by a commercial company: Siemens Healthineers.

“The funder provided support in the form of salaries for authors, but did not have any additional role in the study design, data collection and analysis, decision to publish, or preparation of the manuscript. The specific roles of these authors are articulated in the ‘author contributions’ section.”

Author’s response: There is some misunderstanding. None of the authors are employed by Siemens Healthineers. RV works at a university medical center. The relationship with Siemens Healthineers is limited to institutional research grants and speaker fees, and not employment. We have updated the Competing Interests section to make this distinction clear. We have included this amended Competing Interests section in our cover letter.

I have read the journal's policy and the authors of this manuscript have the following competing interests:

RV declares receiving institutional grants and speaker fees from the following company: Siemens Healthineers.

This does not alter our adherence to PLOS ONE policies on sharing data and materials.

The other authors of this manuscript declare no relationships with any companies, whose products or services may be related to the subject matter of the article.

5. We note that you have indicated that there are restrictions to data sharing for this study. For studies involving human research participant data or other sensitive data, we encourage authors to share de-identified or anonymized data. However, when data cannot be publicly shared for ethical reasons, we allow authors to make their data sets available upon request. For information on unacceptable data access restrictions, please see http://journals.plos.org/plosone/s/data-availability#loc-unacceptable-data-access-restrictions.

Author’s response: In response to point a), we regret we cannot share the data publicly due to privacy concerns and restrictions imposed by the NELSON Data Access Board. We have updated our Data Availability statement to align fully with PLOS One requirements and now also include contact information for the NELSON Data Access Board. In response to point b), due to the restrictions imposed by the NELSON Data Access Board we cannot upload an anonymized data set.

Data cannot be shared publicly because of privacy concerns. Data are available from the NELSON Data Access Board (contact via https://umcgresearchdatacatalogue.nl/all/resources/NELSON) for researchers who meet the criteria for access to confidential data.

6. One of the noted authors is a group or consortium “NELSON-POP study group”. In addition to naming the author group, please list the individual authors and affiliations within this group in the acknowledgments section of your manuscript. Please also indicate clearly a lead author for this group along with a contact email address.

Author’s response: Our apologies for misreporting the name of the consortium in the author list. We have updated the author list to correctly refer to the NELSON-POP consortium. We have also indicated the lead author and added contact information for the lead author.

Changes made in line 11:

NELSON-POP consortium^

Changes made in line 19:

^Membership of the NELSON-POP consortium is provided in the Acknowledgements.

Changes made in line 342-344:

All members of the NELSON-POP consortium:

Lead author: Rozemarijn Vliegenthart, MD, PhD, University Medical Center Groningen, University of Groningen, Groningen, The Netherlands. Email: r.vliegenthart@umcg.nl

7. Please amend either the abstract on the online submission form (via Edit Submission) or the abstract in the manuscript so that they are identical.

Author’s response: We have amended the abstract on the online submission form to align with the abstract in the manuscript.

8. Please upload a new copy of Figures 5, 6, S2, S4 and S5, as the detail is not clear. Please follow the link for more information: https://journals.plos.org/plosone/s/figures

Author’s response: Thank you for bringing these deficiencies to our attention. We have used PLOS’s free figure tool NAAS to re-process the images. We have uploaded these new versions.

Author’s response: We have updated our supporting information files, updated in-text citations, and included a Supporting Information files section at the end of the manuscript. In the Supporting Information files section we have

added titles and captions.

Changes made in line 450-455:

Supporting Information

S1 Appendix. Training of Truncated FOV method’s models.

S2 Fig. Correlation of BMI with SAT. Correlation plots between BMI and SAT for the truncated, compensated, and extended FOV methods.

S3 Appendix. Details on the registration process for compensated FOV.

S4 Appendix. Differences in measurements based on scan kVp.

Author’s response: No reviewer has requested to cite specific previously published works.

Additional Editor Comments:

This study evaluates and compares the repeatability of three automated methods for measuring body composition—specifically skeletal muscle area and subcutaneous adipose tissue area—from low-dose chest CT scans in a longitudinal setting. The manuscript is concise, well-organized, and clearly presented. Although it does not introduce significant technical novelty, it represents a valid investigation that offers meaningful insights for the medical field, supported by a large dataset. That said, I have several technical questions that require clarification:

First, I have concerns regarding the radiodensity discrepancies observed among the three methods. The extended FOV method underwent notably different preprocessing compared to the other two methods. It remains unclear how the authors could confidently claim—and rule out—that these discrepancies were not attributable to preprocessing. In particular, the fact that discrepancies persisted when the extended FOV method was applied to datasets from the other two methods suggests to me that its processing pipeline may introduce systematic errors. As the authors noted, downsampling could indeed lead to segmentation issues.

Author’s response: Our apologies for being unclear. We intended to convey that when the extended FOV preprocessing steps (downsampling from 512x512 to 256x256 and clipping to -150HU to 150HU) were applied as post-processing steps to the scans and segmentations used in and created by the truncated/compensated FOV methods, the HU measurements they provide did not change. That is why we concluded that the preprocessing did not affect the HU measurements. We have clarified this in the discussion.

Changes made in line 276-279:

When the preprocessing steps of the extended FOV method named in a) and b) were applied as post-processing steps to the CT scans and segmentations from the truncated/compensated FOV method, the radiodensity measurements provided by the truncated/compensated FOV method did not change.

Second, regarding the study population, all CT data are from male participants. The rationale for including only males—or whether only male data were available—is not explained. Further details on the population selection criteria are needed.

Author’s response: Unfortunately so few females with short-term follow-up scans were available that it would have introduced unintended variability in our technical evaluation of the effect of FOV cutoff on measurement repeatability. Therefore we selected only men. We have clarified this in the methods.

Changes made in line 90-95:

This subset contained a group of 562 male subjects with two CT scans taken at an interval of 3-4 months. In the NELSON trial, only 13.9% of the available short-term follow-up CT data from University Medical Center Utrecht was from females, inclusion of which would have introduced unintended variability in the technical evaluation of the effect of FOV cutoff of tissues on measurement repeatability.

---

## [Editor Report · Decision Letter 1]

15 Feb 2026

Repeatability of automated body composition measurement on low dose chest CT in male subjects

PONE-D-25-45870R1

Dear Dr. Bunk,

We’re pleased to inform you that your manuscript has been judged scientifically suitable for publication and will be formally accepted for publication once it meets all outstanding technical requirements.

Kind regards,

Zhentian Wang, Ph.D.

Academic Editor

PLOS One

Additional Editor Comments (optional):

The authors have addressed all the reviewers' comments satisfactorily.
---

## [Editor Report · Acceptance letter]

PONE-D-25-45870R1

PLOS One

Dear Dr. Bunk,

I'm pleased to inform you that your manuscript has been deemed suitable for publication in PLOS One. Congratulations! Your manuscript is now being handed over to our production team.

Kind regards,

on behalf of

Prof. Zhentian Wang

Academic Editor

PLOS One